# Proposal of the Implementation Theory Selection Model and exemplar application in fall injury prevention

**Alexandra M. B. Korall[1,2], Helen Chong[3], Vicki Komisar[3,4], Dawn C. Mackey[3], Masood Khan[2], Femke Hoekstra[5], Susan G. Brown[6], Pauli Gardner[7], Christine Hames[8], Andrew C. Laing[6,9], Kathryn M. Sibley[1,2]***

**1** George & Fay Yee Centre for Healthcare Innovation, Winnipeg, MB, Canada, **2** Department of Community Health Sciences, Rady Faculty of Health Sciences, University of Manitoba, Winnipeg, MB, Canada, **3** Department of Biomedical Physiology and Kinesiology, Simon Fraser University, Burnaby, BC, Canada, **4** School of Engineering, University of British Columbia, Kelowna, BC, Canada, **5** School of Health and Exercise Sciences, University of British Columbia, Okanogan, BC, Canada, **6** Schlegel-University of Waterloo Research Institute for Aging, Waterloo, ON, Canada, **7** Department of Health Sciences, Brock University, St. Catharines, ON, Canada, **8** Program for Active Living, Schlegel Villages, Kitchener, ON, Canada, **9** Department of Kinesiology and Health Sciences, University of Waterloo, Waterloo, ON, Canada

* Kathryn.Sibley@umanitoba.ca

**Data Availability Statement:** All relevant data are within the manuscript and its Supporting Information files.

**Funding:** This work was supported by the AGE-WELL (Aging Gracefully Across Environments

## Abstract

### Introduction

The use of theories, models and/or frameworks (TMFs) in implementation research and practice is essential for developing useful and testable implementation strategies. Recommendations and tools exist to aid implementation groups in selecting TMFs, but they do not explicitly outline a systematic method for identifying and selecting TMFs. This paper aimed to (1) propose a systematic consensus-based method to select TMFs to support implementation processes, and to (2) demonstrate the use of this novel method in the context of researching the implementation of hip protectors for fracture prevention in long-term care (LTC).

### Materials & methods

We developed a systematic, consensus-based method for selecting TMFs, referred to as the **I**mplementation **T**heory **S**election **M**odel (ITSM). The ITSM comprises five steps: (1) identify potentially relevant TMFs; (2) narrow the pool of TMFs; (3) appraise the relevance of eligible TMFs; (4) prioritize a short-list of TMFs for further, in-depth consideration; and (5) select TMFs through consensus with investigators and research user partners. We operationalized each step of the ITSM through a project investigating determinants of hip protector use and organizational readiness for implementation in a LTC organization in Ontario, Canada.

### Results

Using the ITSM in our case example, we identified 66 TMFs (Step 1). Of these, 23 met our eligibility criteria (Step 2) and were appraised twice, by five appraisers (Step 3). Six TMFs

Using Technology to support Wellness, Engagement and Long Life) Network of Centres of Excellence (award numbers AWCAT-2019-143 and AWCRP-2020-04). The funding source had no role in the design and conduct of the study; collection, management, analysis, and interpretation of data; preparation, review or approval of the manuscript; and decision to submit the manuscript for publication.

**Competing interests:** The authors declare that they have no competing interests.

(Step 4) advanced to the consensus meeting, which was attended by nine investigators and three research users, including two organizational partners and one older adult. Three rounds of voting yielded a tie between the TMFs the group felt would be most appropriate. Research users from our partner LTC organization made the final selection preferring the combination of the Practical, Robust Implementation and Sustainability Model and Consolidated Framework for Implementation Research (Step 5).

## Conclusions

The ITSM offers a step-by-step guide for implementation groups to adopt a rigorous, transparent and reproducible method for TMF selection. Although we have demonstrated the feasibility of operationalizing each step of the ITSM in our case example, continued research is needed to evaluate and refine the ITSM to ensure it is appropriate for a wide variety of implementation contexts.

## Introduction

Implementation scientists have long advocated for the meaningful use of theories, models and/or frameworks (TMFs) in implementation research and practice [1, 2]. The use of TMFs is essential for developing useful and testable implementation strategies [2], promoting generalizability of findings across diverse settings through shared language and understanding [3], and facilitating knowledge synthesis and the accumulation of an evidence-base through standardized construct definitions and measurement [4]. Most implementation research, however, is uninformed, or ineffectively informed, by TMFs [5–7]. TMFs are often underused, misused or superficially used [8].

Implementation TMFs have varied aims. Some are useful for guiding the process of implementation, some for understanding and/or explaining what influences implementation outcomes, and others for evaluating implementation outcomes [9]. However, TMFs with a similar aim often contain different constructs, terminologies, and definitions [10]. The choice of TMFs can affect the process and outcomes of implementation. No single TMF, or combination of TMFs, is considered superior over others in its suitability to inform all implementation research, and the question of "*Which TMF(s) should I use*?" has persisted for decades [2]. The complexity of selecting implementation TMFs is amplified by the number of existing TMFs to choose from [8]. For example, a scoping review identified 159 TMFs that have been used, many only once (60%), in published implementation research for preventing and managing cancer and chronic diseases [11].

Implementation teams report that they consider a variety of criteria when selecting TMFs [8], and involving organizational and community partners in research decision-making, which includes TMF selection, has been proposed as a critical component of implementation [12]. However, there is little agreement as to which criteria are most important, and the ultimate selection of TMFs is often determined primarily by convenience or familiarity [8]. Recommendations and tools exist to help choose TMFs. Moulin et al. (2020) recommend considering the purpose, analytic level, degree of specificity and orientation of TMFs [4]. Scientists from Canada, the United Kingdom, and the United States developed the Theory Comparison and Selection Tool (T-CaST), which contains 16 criteria across four categories (applicability, usability, testability, and acceptability) that can be used to justify selection of a

TMF for a given project [13]. While these resources offer valuable guidance, they do not operationalize a systematic method for identifying and selecting TMFs, which can be effectively replicated across implementation efforts and contexts.

This paper aimed to (1) propose a systematic, consensus-based method to identify and select TMFs to support implementation processes, and to (2) demonstrate the use of this novel method in the context of researching the implementation of hip protectors for fracture prevention in long-term care (LTC).

## Materials and methods

### Implementation context

Globally, approximately 1.6 million older people sustain a hip fracture each year [14]. Falls cause 95% of hip fractures in older people [15]. The consequences of hip fracture can be especially severe for LTC residents [16–19]. Older people who fracture their hip experience heightened morbidity and mortality [20–24], reduced quality of life [25, 26], and require greater health care use (e.g., emergency department visits, surgery, physician contact) [27].

Hip protectors are one strategy for managing falls and preventing hip fractures in LTC. They consist of hard shields or soft pads sewn or inserted into elasticized pockets of garments or undergarments, which cover the skin over the lateral aspects of the proximal femur. Hip protectors are designed to reduce the risk of hip fracture from falls by absorbing and/or shunting the impact energy away from the proximal femur [28]. Although the efficacy of commercial models of hip protectors varies [29], certain models likely reduce the risk for hip fractures in LTC when used as intended [30, 31]. Hip protectors have been recommended for older people in LTC, especially for those who are mobile and at high risk of falls [32, 33], but they only protect against hip fracture if worn during the fall. Research suggests hip protectors are not used consistently in LTC, which limits their clinical effectiveness [34]. There is a need for implementation research on how to enhance uptake and sustain the use of hip protectors in LTC [35].

### Philosophical foundation and theoretical underpinnings

Our philosophical approach to this work was rooted in pragmatism, a paradigm that aims to use research findings to solve practical real-world problems [36]. We aimed to develop and apply a systematic method to identify and select TMFs in the context of researching the implementation of hip protectors for fracture prevention in LTC. Consistent with an Integrated Knowledge Translation (IKT) model of collaborative research, our method considers the views of research users (those able to use research findings to make decisions) when reaching consensus on the final selection of TMFs [37, 38]. The goal is to receive and incorporate the feedback and advice of research users when deciding on the final choice of TMFs. Our systematic method for TMF identification and selection was guided by the James Lind Alliance (JLA) process (*https://www.jla.nihr.ac.uk/jla-guidebook/*). The JLA brings patients, clinicians, and care partners together in Priority Setting Partnerships to reach consensus on the most important unanswered questions for future research.

### Research ethics statement

The individuals involved in the theory selection process were not human research participants, but instead were partners on the research team (level of 'consult' on the IAP2 Spectrum of Patient and Researcher Engagement in Health Research). Given the collaborative decision making role of these partners in the research process, research ethics board (REB) approval

was not sought. This practice was consistent with the JLA guidance that views such decision making work as service evaluation and development (*https://www.jla.nihr.ac.uk/jla-guidebook/*).

## Overview of the Implementation Theory Selection Model (ITSM)

We developed a systematic, consensus-based method for identifying and selecting TMFs, referred to as the **I**mplementation **T**heory **S**election **M**odel (ITSM). The ITSM comprises five steps: (step 1) identify potentially relevant TMFs; (step 2) narrow the pool of TMFs to only those with an appropriate aim; (step 3) appraise the relevance of eligible TMFs; (step 4) prioritize a short-list of TMFs for in-depth consideration; and (step 5) select TMFs through consensus with investigators and research user partners (Fig 1). Table 1 summarizes each step of the ITSM.

The ITSM was informed by the JLA method for priority setting (*https://www.jla.nihr.ac.uk/jla-guidebook/*). The JLA method includes steps for gathering data on potential research questions, evidence checking, interim priority setting, and the final prioritization of research questions through a workshop. AMBK and KMS proposed a first iteration of the ITSM's five steps, which were revised through consultation with our academic and organizational team members. Once the five steps of the ITSM were finalized, AMBK and KMS proposed the sub-steps to carry out each step of the ITSM in our case example on hip protector implementation, which was revised iteratively through discussions with our research team as the process evolved. In the sections below, we describe and provide examples of how we operationalized each step of the ITSM in our case example on hip protector implementation to illustrate the feasibility and potential value of the ITSM. We encourage ITSM users to work collaboratively with research user partners to develop the specific sub-steps they will use to operationalize the ITSM's five steps which are feasible given their access to available resources.

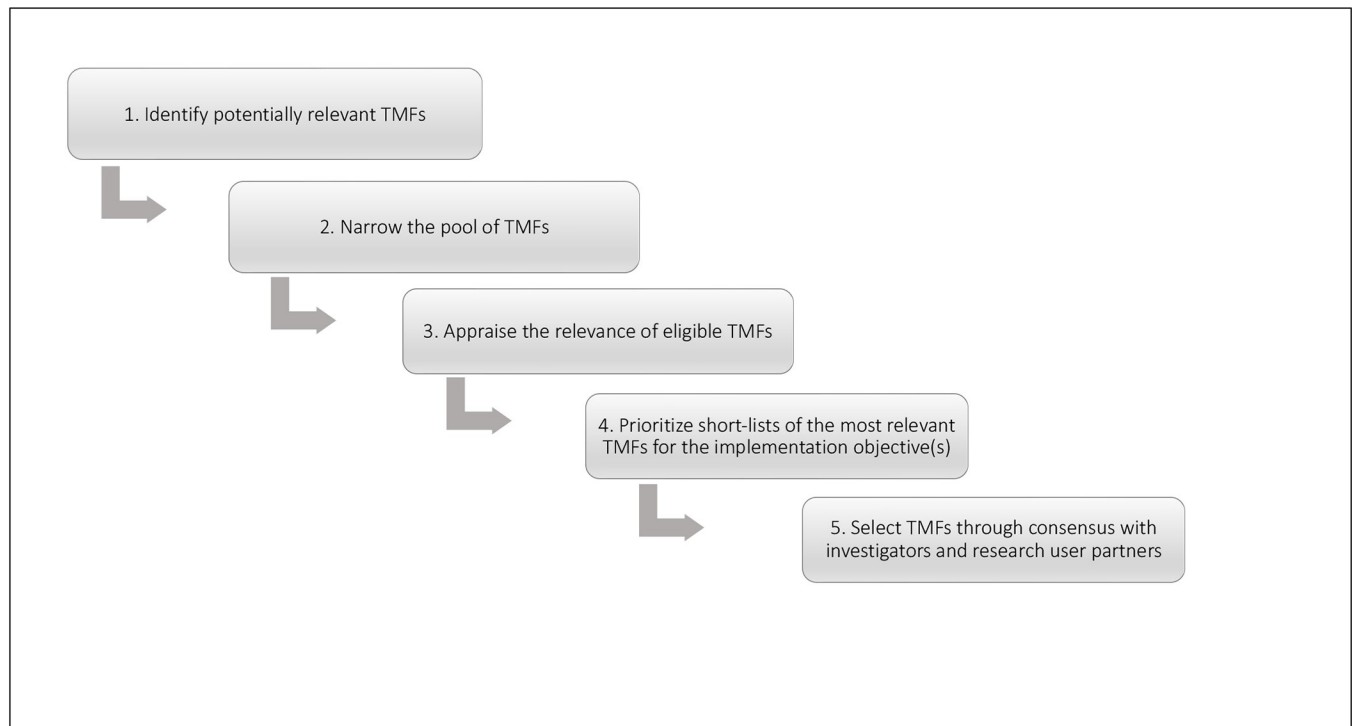

**Fig 1. Illustration of the Implementation Theory Selection Model (ITSM).** TMF = Theory, Model and/or Framework.

**Table 1. Steps of the Implementation Theory Selection Model (ITSM).**

| Step | Goal | Description |
|---|---|---|
| 1 | Identify potentially relevant TMFs | • Search multiple sources to identify a pool of potentially relevant TMFs, relying on existing works to expedite the process whenever possible |
| | | • Example sources include: database of Dissemination & Implementation (D&I) Models in Health Research and Practice (http://www.dissemination-implementation.org); published reviews and academic articles of implementation TMFs, ideally including some conducted with similar target populations, interventions or settings |
| 2 | Narrow the pool of TMFs | • At a minimum, narrow the pool of TMFs identified to only those that have an appropriate aim or purpose, depending on implementation context |
| | | • Consider narrowing the pool further based on the socio-ecological level(s) included within the TMF, the inclusion and depth of analysis or operationalization of specific implementation constructs, and/or the orientation of the TMF based on the type of intervention and setting |
| | | • Define eligibility criteria, based on aim, etc. |
| | | • Retrieve at least one reference (article, website, etc.) per TMF |
| | | • Screen the references, ideally by at least two reviewers, to determine whether each TMF meets the eligibility criteria |
| 3 | Appraise the relevance of eligible TMFs | • Process spans the recruitment of appraisers, (random, if possible) assignment of TMFs to appraisers, compilation and review of references, appraisal of each TMF |
| | | • Recommend using the Theory Comparison and Selection Tool (T-CaST) to guide the appraisal of TMFs |
| | | • Ideally, each TMF appraised by at least two appraisers |
| 4 | Prioritize short-lists of the most relevant TMFs for the implementation objective(s) | • Specify a manageable number of TMFs (we recommend no more than six) for further, in-depth consideration |
| | | • Use the results of TMF appraisals to rank TMFs from most to least relevant for each implementation objective |
| 5 | Select TMFs through consensus with investigators and research user partners | • Use formal methods of consensus, such as an adapted Nominal Group Technique (NGT) or Delphi Technique, with a group of investigators and research user partners to reach consensus on the final choice of TMF(s) |

TMF = Theory, Model and/or Framework.

## Step 1: Identify a pool of potentially relevant TMFs

The first step of the ITSM requires the use of multiple search strategies to identify a pool of potentially relevant TMFs, relying on existing works to expedite the process whenever possible. A recommended source is the database of Dissemination & Implementation (D&I) Models in Health Research and Practice (http://www.dissemination-implementation.org). This database can be filtered using pre-specified search criteria related to the purpose of the TMF (e.g., "Implementation"), socio-ecological level of influence (e.g., "Individual"), field of origin (e.g., "Nursing"), and the presence of specific constructs (e.g., "Adoption"). Complementary sources may include published reviews and academic articles identifying TMFs used in implementation research and practice, ideally including some conducted with similar target populations, interventions or settings, as these characteristics are relatively easy to assess. In general, we advise against being overly prescriptive when developing a list of candidate TMFs, as the association between characteristics of a TMF and its ultimate relevance will be unclear at this step.

**Case example.** In October 2019, we searched the database of Dissemination & Implementation (D&I) Models in Health Research and Practice (http://www.dissemination-implementation. org) using the pre-specified search criteria of "Implementation" (purpose of TMF), "Individual" (socio-ecological level) and "Organization" (socio-ecological level). To complement this search, we retrieved all TMFs identified in a scoping review of implementation research on cancer and chronic disease management and prevention, which were cited at least 10 times [11], along with the most used TMFs identified in a survey of implementation scientists [8]. Our decision to include TMFs with ten or more citations from the scoping review was pragmatic; our goal was to ensure we weren't missing TMFs that are commonly used in a related field of interest. Finally, we retrieved all TMFs identified in a literature review of implementation research on the prevention and management of falls in older adults [39]. We elected to look beyond this review of implementation research on falls in older adults to compile a comprehensive list of candidate TMFs.

## Step 2: Narrow the pool of TMFs

Implementation TMFs have been categorized into different categories based on their aim or purpose. These include process models, determinant frameworks, classic theories, implementation theories, and evaluation frameworks [9]. Process models describe or guide the implementation process [9]. Determinant frameworks, classic theories and implementation theories specify factors that impact implementation outcomes, describe mechanisms of change, and/or explain certain aspects of implementation [9]. Evaluation frameworks provide a structure for evaluating the success or failure of implementation [9]. The second step of the ITSM requires that the pool of TMFs identified in step 1 is narrowed, at a minimum, to only those that have an appropriate aim or purpose. The pool could be narrowed further based on the socio-ecological level(s) included within the TMF, the inclusion and depth of analysis or operationalization of specific implementation constructs, and/or the orientation of the TMF based on the type of intervention and setting [4]. Once the appropriate criteria have been identified, all TMFs should be screened using a systematic process to ensure they fit this criteria. This requires the retrieval of at least one reference for each TMF. As it can be somewhat challenging to make decisions regarding the eligibility of TMFs, ideally, at least two reviewers should independently screen each TMF, and discrepancies should be resolved through discussion or a third independent reviewer. When a complete dual review process is not feasible, single screening should be conducted only by an experienced reviewer [40].

**Case example.** Our case example had two implementation objectives: to (1) investigate determinants of hip protector use; and (2) assess organizational readiness for implementation of hip protectors. We narrowed our search to determinant frameworks and implementation theories since their aims most closely aligned with our objectives. We excluded TMFs when we could not retrieve at least one reference for the TMF. Two reviewers (AT and AMBK) independently screened a sample of ten TMFs, and then met to compare decisions and discuss and resolve discrepancies. A single reviewer (AT) then screened all remaining TMFs for eligibility. To ensure TMFs were not missed by the first reviewer, the subset of TMFs deemed ineligible or unclear by AT were screened by an experienced reviewer (AMBK) to verify their eligibility.

## Step 3: Appraise the relevance of TMFs

The third step of the ITSM appraises the relevance of each eligible TMF. This process spans the recruitment of appraisers, the (random, if possible) assignment of TMFs to appraisers, the compilation and review of references (e.g., academic articles, websites, questionnaires, etc.), and the appraisal of TMFs. We recommend using the T-CaST to assess the relevance of each

TMF for the implementation objective(s) [13]. We adapted the original 16-item version of the T-CaST for Implementation Researchers [13] to create an Appraisal Form that can be used to appraise the relevance of a given TMF (**SI File**). A first iteration of the Appraisal Form was piloted by five team members, on two TMFs. After independently appraising both TMFs, the team met remotely via a two-hour online meeting to discuss and refine the Appraisal Form. Table 2 describes and provides rationales for modifications made to original T-CaST items. The final Appraisal Form contains 12 items probing the usability (5 items), applicability (4 items), and testability (3 items) of eligible TMFs. Appraisers indicate the extent to which they disagree or agree with each statement by responding on a 5-point Likert scale, ranging from 1 (strongly disagree) to 5 (strongly agree). The Appraisal Form also contains items probing the overall perceived fit of the TMF to each implementation objective. Appraisers rate the fit of the TMF on a 3-point Likert scale, ranging from 0 (poor fit) to 2 (good fit). Ideally, each TMF should be appraised by at least two appraisers.

**Case example.**   We invited six investigators and one research user from our partner LTC organization to participate in the appraisal of TMFs. We asked each appraiser to self-identify as either a novice or advanced appraiser. We defined a novice appraiser as having little to no familiarity with implementation TMFs, and an advanced appraiser as having some to substantial familiarity with implementation TMFs.

We randomly assigned TMFs to appraisers using block randomization, ensuring that each TMF was appraised by one advanced and one novice appraiser. AMBK emailed appraisers a folder containing high-priority (recommended) and low-priority (optional) references for each TMF. We prioritized references that described the initial development and/or testing of the TMF, subsequent iterations of the TMF, along with example applications in the LTC setting. Based on feedback from pilot testing, no more than four references were categorized as high-priority for each TMF. We asked appraisers to read the high-priority references, at a minimum, before completing the Appraisal Form. Appraisers were also permitted to search for and retrieve additional references for a given TMF, as needed.

## Step 4: Prioritize short-lists of the most relevant TMFs for the implementation objective(s)

Once all TMFs have been appraised, Step 4 of the ITSM uses data from Step 3 to prioritize short-lists of the most relevant TMFs for the implementation objective(s). The first step is to specify a manageable number of TMFs for further, in-depth consideration. This will vary across teams, but we advise shortlisting no more than six TMFs to advance to Step 5. Then, using the data collected during the appraisal process, TMFs should be ranked by a member of the research team from most to least relevant for each implementation objective.

**Case example.** We short-listed three TMFs per implementation objective to advance to step 5. We calculated the mean fit of each TMF per implementation objective by averaging 3-point Likert-scale responses across appraisers. For each TMF, we also calculated a mean overall T-CaST score. This was done by: (1) calculating an individual appraiser's mean usability, testability, and applicability scores; (2) summing an individual appraiser's mean usability, applicability, and testability scores to derive an overall T-CaST score per appraiser (value range: 3–15); (4) averaging overall T-CaST scores across appraisers. Mean fit scores varied by implementation objective, whereas overall T-CaST scores did not. We ranked TMFs according to their mean fit score for a given implementation objective, and then by mean overall T-CaST score. Data were analyzed using JMP PRO version 14 (SAS Institute Inc, Cary, NC).

**Table 2. Items contained in the Theory Comparison and Selection Tool (T-CaST) for implementation researchers, items included in our appraisal form, and reasons for modifying or excluding T-CaST items.**

| Original T-CaST item | Item included in Appraisal Form | Rational for modification/exclusion |
|---|---|---|
| *Usability* | | |
| 1 TMF includes relevant constructs (e.g., self-efficacy; climate) | TMF includes relevant constructs[1] (e.g., readiness; any other constructs you perceive are relevant). [1]Note that the term 'constructs' is used broadly to refer to the types of variables (also known as classes or domains), as well as specific variables within a category, class or domain that are contained in the TMF. | **Modification**: Appraisers were initially constrained by examples contained in original item and requested clarification as to what is meant by the term 'constructs' |
| 2 Key stakeholders (e.g., researchers; clinicians; funders) are able to understand, apply, and operationalize TMF. | N/A | **Exclusion**: Appraisers were not confident in their ability to anticipate whether research users would be able to understand, apply and operationalize TMF |
| 3 TMF has a clear and useful figure depicting included constructs and relationships among them. | TMF has a clear and useful figure or table depicting included constructs and relationships among them. | **Modification**: Appraisers expressed that a table could be equally useful as a figure |
| 4 TMF provides a step-by-step approach for applying it. | TMF provides a step-by-step approach for applying it. | N/A |
| 5 TMF provides methods for promoting implementation in practice. | TMF provides methods for promoting implementation in practice. | N/A |
| 6 TMF provides an explanation of how included constructs influence implementation and/or each other. | TMF provides an explanation of how included constructs influence implementation and/or each other. | N/A |
| *Testability* | | |
| 7 TMF proposes testable hypotheses. | TMF proposes testable hypotheses. | N/A |
| 8 TMF includes meaningful, face-valid explanations of proposed relationships. | TMF includes meaningful, face-valid explanations of proposed relationships and constructs. | **Modification**: Appraisers expressed that it was important for TMFs to also include meaningful, face-valid explanations of constructs contained in the TMF |
| 9 TMF contributes to an evidence base and/or theory development because it has been used in empirical studies. | TMF contributes to an evidence base and/or theory development because it has been used in empirical studies. | N/A |
| *Applicability* | | |
| 10 TMF focuses on a relevant implementation outcome (e.g., fidelity; acceptability). | TMF focuses on a relevant implementation outcome (e.g., acceptability; adoption; appropriateness; feasibility; fidelity; implementation cost; penetration; sustainability). | **Modification**: Appraisers were initially constrained by examples contained in original item |
| 11 A particular method (e.g., interviews; surveys; focus groups; chart review) can be used with TMF. | N/A | **Exclusion**: Appraisers expressed that investigators have expertise in mutliple research methods and compatability of TMF with a particular method was inconsequential |
| 12 TMF addresses a relevant analytic level (e.g., individual; organizational; community). | TMF addresses one or more relevant analytic levels (e.g., individual; organization). | **Modification**: Appraisers expressed a desire for specific examples of relevant analytic levels and clarification that a single TMF need not address all relevant analytic levels |
| 13 TMF has been used in a relevant population (e.g., children; adults with serious mental illness) and/or conditions (e.g., attention deficit hyperactivity disorder; cancer). | TMF is generalizable to our context. | **Modification**: Appraisers expressed that it was not feasible to assess what populations and/or conditions a TMF has been used in without doing a review of the literature |
| 14 TMF is generalizable to other disciplines (e.g., education; health services; social work), settings (e.g., schools; hospitals; community-based organizations), and/or populations (e.g., children; adults with serious mental illness). | TMF is generalizable to other disciplines and/or contexts. | **Modification**: Appraisers were initially constrained by the examples contained in the original item and expressed a desire to simplify the statement |
| *Acceptability* | | |
| 15 TMF is familiar to key stakeholders (e.g., researchers; scholars; clinicians; funders). | N/A | **Exclusion**: Appraisers were not confident in their ability to anticipate whether stakeholders are familiar with a given TMF |

*(Continued)*

**Table 2.** (Continued)

| Original T-CaST item | Item included in Appraisal Form | Rational for modification/exclusion |
|---|---|---|
| 16 | TMF comes from a particular discipline (e.g., education; health services; social work). | N/A | **Exclusion**: Appraisers expressed the discipline of origin of a TMF was inconsequential |

TMF = Theory, Model and/or Framework.

## Step 5: Select TMFs through consensus

Step 5 of the ITSM reaches consensus on the final choice of TMF(s). Consensus should be reached jointly by a team of investigators and research user partners using formal methods of consensus, such as an adapted Nominal Group Technique (NGT) or Delphi Technique. The NGT uses face-to-face communication, and includes six stages: (1) formulation and presentation of a question; (2) silent idea generation; (3) round robin feedback and recording of ideas; (4) group discussion; (5) individual voting or polling; (6) tallying of votes, feedback of results, further group discussion (optional) and re-voting (optional) [41]. The Delphi Technique uses a series of questionnaires to reach consensus, often spanning weeks to months [42]. In general, the NGT is better suited when consensus is needed quickly (i.e., one day) and from a smaller group (less than 15) compared to the Delphi Technique [42].

**Case example.** We invited all investigators and research user partners (n = 21) to attend a virtual, two-hour consensus meeting. We sent a formal invitation and reminder emails from the email account of our research manager, HC. Ten days before the consensus meeting, we distributed summaries that we created of the short-listed TMFs, along with examples of guiding questions (developed by AMBK and KMS) to facilitate participation in the consensus meeting. Summaries were two pages, and included a summary statement of the TMF, a graphical illustration of the TMF, an overview of key constructs contained in the TMF, the year the TMF was first published, the estimated number of times the TMF had been cited (abstracted from https://dissemination-implementation.org or Google Scholar), example applications of the TMF (applications in LTC setting and similar health conditions prioritized), and a summary of the appraisal scores, including mean fit per implementation objective, along with mean usability, applicability, and testability scores (see **S2 File** for an example). We also emailed references for each TMF under consideration, but these readings were optional. We encouraged attendees to send any questions via email and/or to arrange a time to chat virtually with AMBK prior to the consensus meeting. The day before the consensus meeting, we emailed attendees a copy of the meeting agenda, a summary of the consensus process, and guidelines for discussion.

The consensus meeting was hosted online using virtual meeting software (Zoom) and was recorded. We hired an independent postdoctoral fellow (FH) to facilitate, who had experience with implementation TMFs (the Delbecq technique [41]). The facilitator began by reviewing the agenda (**S3 File**) and establishing ground rules for discussion. We used an adapted NGT [41] with three rounds of consensus. We used NGT as we wanted to reach consensus quickly, while still considering everyone's opinions. Table 3 lists the questions asked in each round, which were shared with attendees before the meeting.

Round 1 and Round 2 were unconventional, parallel rounds, as the outcome of the first did not influence the outcome of the second. The goal of the first round was to eliminate one of the top three TMFs for our first implementation objective to investigate factors influencing use of hip protectors. The goal of the second round was to eliminate one of the top three TMFs for our second objective to assess organizational readiness for implementation. The facilitator

**Table 3. Questions asked during each round of consensus in our case example.**

| Round | Question |
| --- | --- |
| 1 | Which of the following TMFs is least relevant for guiding our initial investigation of barriers and facilitators to the use of hip protectors in [our partner] long-term care homes? |
| 2 | Which of the following TMFs is least relevant for guiding our initial exploration of organizational readiness to implement, monitor and evaluate strategies to increase use of hip protectors in [our partner] long-term care homes? |
| 3 | What is your preferred combination of TMFs to collectively underpin our objectives? |

TMF = Theory, Model and/or Framework.

began Round 1 and Round 2 by presenting a question to the group (Table 3). AMBK then presented a brief overview (1 minute per TMF) of the top three TMFs for each implementation objective, after which attendees could ask clarifying questions about the TMFs under consideration. As we asked attendees to come to the meeting prepared to share their views to the questions posed in the first two rounds, we did not allocate time for silent idea generation. Then, each participant was given an opportunity to contribute their views about which TMF was least relevant for either objective, depending on the round, and why. A note taker summarized each participant's views in the chat window. A discussion followed exploring areas of agreement and divergence and the clarification of any uncertainties. Attendees indicated if they wanted to make a comment by clicking a virtual hand raise button and were provided one minute to make comments. After the discussions, attendees were asked to vote for the least relevant TMF for each implementation objective by clicking on links posted in the chat window to online questionnaires hosted by Survey Monkey.

Fig 2 illustrates the voting process for Round 1 and Round 2. If the first poll did not yield a majority (more than 50%) of votes, then the two TMFs with the most votes to be eliminated advanced to a second poll to declare which TMF would be eliminated [43]. We broke ties using mean overall T-CaST scores derived from TMF appraisals. In the event of a three-way tie in the first poll, the two TMFs with the lowest mean overall T-CaST scores advanced to a second poll. In the event of a two-way tie in the second poll, the TMF with the lowest mean overall T-CaST score was eliminated. We tallied and presented the voting results at the end of each round.

The third round aimed to agree upon a final selection of TMFs to collectively underpin our implementation objectives. The facilitator instructed attendees that combinations of TMFs must include at least one of the remaining two TMFs from round 1, and at least one from round 2. We allocated five minutes for silent idea generation. Then, each participant was given an opportunity to contribute their views, and a summary of their comments was posted in the chat window. This was followed by a discussion period. We did not use live polling for voting. Votes could be submitted up to one week after the meeting via an online questionnaire administered via Survey Monkey, with links posted to the chat window and emailed to each attendee. We asked attendees to identify their first, second and third choice combinations of TMFs. We assigned three points for every first-choice vote, two points for every second-choice vote, and one point for every third-choice vote. We summed the total points assigned to each possible combination of TMFs (nine possible combinations) and ordered these from most to least points. The results were tallied and presented to the group in a follow-up meeting. The combination of TMFs with the most points was selected. In the event of a tie, we asked research users from our partner LTC organization (CH and SB) to decide which of the highest ranked combinations of TMFs they preferred. To inform their decision-making, we provided our partners

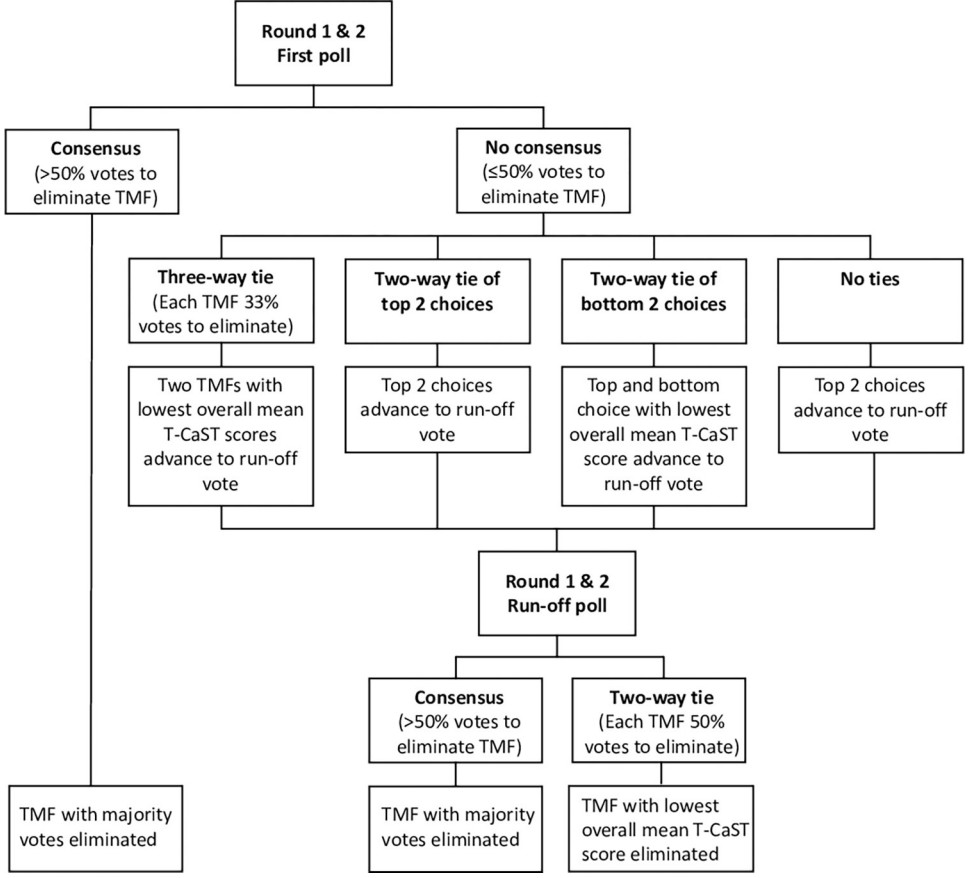

**Fig 2. Elimination process for the first two parallel rounds of consensus in our case example.** TMF = Theory, Model and/or Framework.

with: (1) summaries of the TMFs still under consideration; (2) references for the TMFs still under consideration; (3) voting results and a summary of the group discussions from all three rounds of the consensus meeting.

## Results

### Case example

The flow of TMFs through the ITSM in our case example is shown in **Fig 3**.

### Step 1: Identify a pool of potentially relevant TMFs

After removing duplicates, we identified 66 potentially relevant TMFs.

### Step 2: Narrow the pool of TMFs

Of the 66 potentially relevant TMFs, 23 TMFs met our eligibility criteria, including 9 determinant frameworks and 14 implementation theories. Reasons for exclusion were as follows: could not retrieve at least one seminal source (n = 1); classic theory (n = 15); process model (n = 19), evaluation framework (n = 3) or other aim of TMF (n = 5). The names and categorized aims of included and excluded TMFs are listed in **S4** and **S5 Files**, respectively.

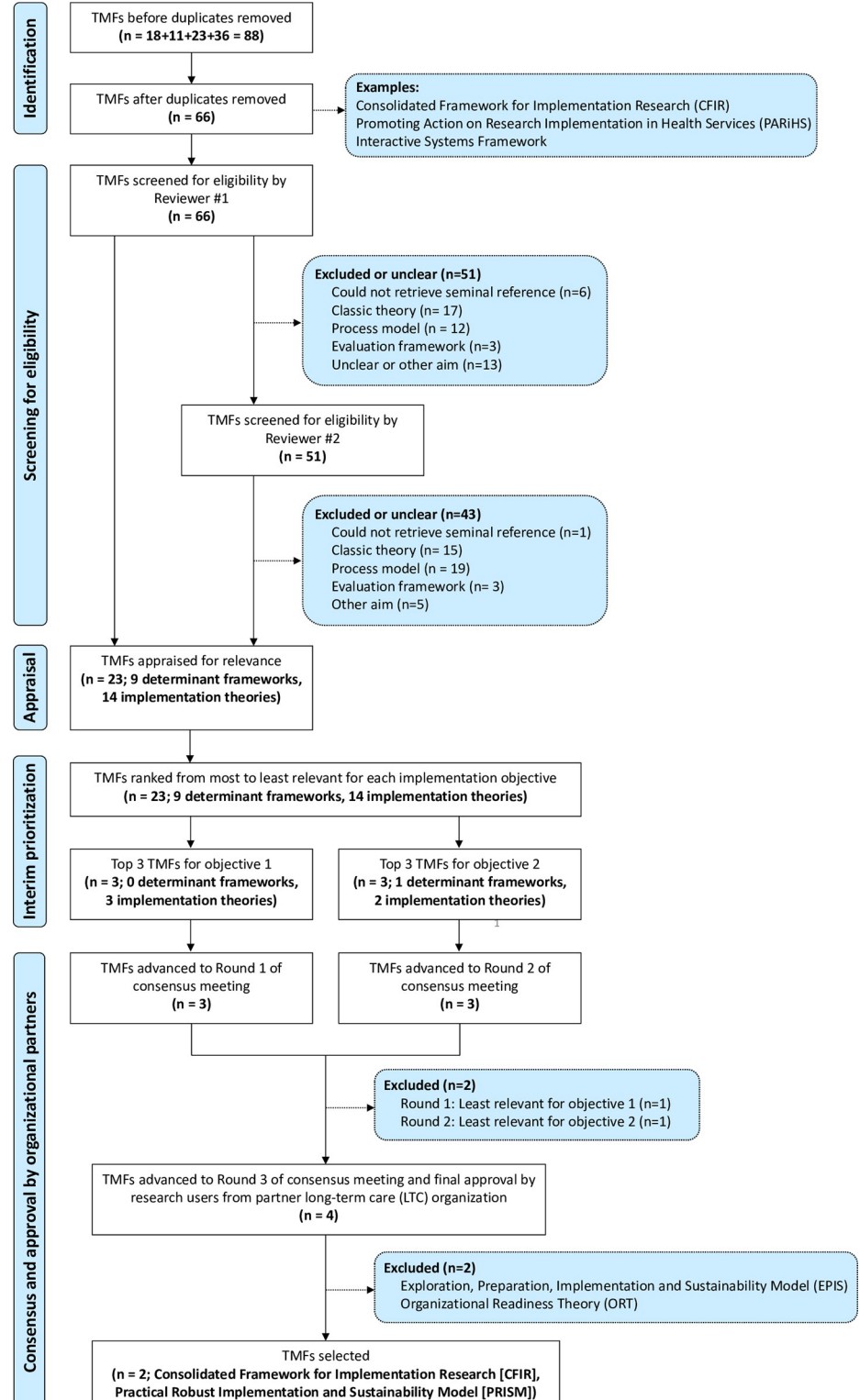

**Fig 3. Flow of TMFs through the Implementation Theory Selection Model (ITSM) in our case example.**
TMF = Theory, Model and/or Framework.

## Step 3: Appraise the relevance of TMFs

Five investigators appraised the 23 eligible TMFs (**S6 File**). Two self-identified as advanced appraisers, while three self-identified as novice appraisers. Appraisers had similar qualifications in kinesiology and engineering. The highest degree earned was a Ph.D. for four appraisers, and a M.Sc. for one appraiser. The appraisal process spanned two months.

Mean fit scores ranged from 0.5 to 2 (out of 2), and from 0 to 2 (out of 2) for objectives 1 and 2, respectively. Four TMFs had a mean fit score of 2 ("good fit") for objective 1, and two TMFs had a mean fit score of 2 ("good fit") for objective 2. Mean usability, testability, and applicability scores ranged from 1.8–4.4 (out of 5), 1.8–4.7 (out of 5), and 2.4–4.9 (out of 5) across TMFs, respectively.

## Step 4: Prioritize short-lists of the most relevant TMFs for the implementation objective(s)

Table 4 shows the ranked order of TMFs by relevance, from most (rank = 1) to least (rank = 23) relevant, for each implementation objective. The Top 3 TMFs for objective 1 were the Normalisation Process Theory (NPT) [44], the Exploration, Preparation, Implementation, Sustainment (EPIS) Model [45, 46] and the Practical, Robust Implementation and Sustainability Model (PRISM) [47]. The Top 3 TMFs for objective 2 were the Organizational Readiness

**Table 4. Ranked order of the perceived relevance of theories, models and/or frameworks for the implementation objectives in our case example.**

| Theory, Model or Framework | Rank | |
|---|---|---|
| | Objective 1 | Objective 2 |
| Normalization Process Theory (NPT) | 1[a] | 4 |
| Exploration, Preparation, Implementation and Sustainability (EPIS) Model | 2[a] | 5 |
| Practical, Robust Implementation and Sustainability Model (PRISM) | 3[a] | 6 |
| Theoretical Domains Framework (TDF) | 4 | 19 |
| Precede-Proceed Model | 5 | 3[a] |
| Critical Realism & the Arts Research Utilization Model (CRARUM) | 6 | 8 |
| Consolidated Framework for Implementation Research (CFIR) | 7 | 2[a] |
| Promoting Action on Research Implementation in Health Services (PARiHS) framework | 8 | 7 |
| Dissemination of Evidence-based Interventions to Prevent Obesity | 9 | 17 |
| Organizational Readiness Theory (ORT) | 10 | 1[a] |
| Sticky Knowledge | 11 | 14 |
| Push-Pull Capacity Model | 12 | 15 |
| Implementation Drivers Framework | 13 | 16 |
| Behaviour Change Wheel (BCW) | 14 | 20 |
| Research Development Dissemination and Utilization Framework | 15 | 11 |
| Implementation of Change in Health Care | 16 | 23 |
| Active Implementation Framework | 17 | 9 |
| Implementation Effectiveness Model / Organizational Theory of Implementation of Innovations | 18 | 10 |
| Conceptual Model of Implementation Research | 19 | 21 |
| Interactive Systems Framework for Dissemination and Implementation | 20 | 12 |
| Knowledge Transfer and Exchange | 21 | 13 |
| Real World Dissemination | 22 | 18 |
| Social Marketing Framework | 23 | 22 |

[a]Top 3 for each objective short-listed for consideration in the consensus meeting.

Theory [48], the Consolidated Framework for Implementation Research (CFIR) [10] and the Precede-Proceed Model [49].

## Step 5: Select TMFs through consensus

Nine investigators and three research users, including two organizational partners and one older adult member of our community advisory board, attended the consensus meeting. Three investigators elected not to participate in group discussions or voting.

Nine attendees voted in the first poll for Round 1. Of these, eight (89%) voted to eliminate the Normalization Process Theory, and one (11%) voted to eliminate PRISM. As the first poll of Round 1 yielded a majority of votes, NPT was eliminated.

Eight attendees voted in the first poll for Round 2. Of these, seven (88%) voted to eliminate Precede-Proceed, and one (13%) voted to eliminate Organizational Readiness Theory. As the first poll of Round 2 yielded a majority of votes, Precede-Proceed was eliminated.

The four TMFs that advanced to Round 3 were EPIS, PRISM, Organizational Readiness Theory, and CFIR. Six attendees voted in Round 3. Seven combinations of TMFs received at least one vote (Table 5). Combinations had to contain at least one of the remaining TMFs from Round 1 (EPIS and PRISM), and at least one of the remaining TMFs from Round 2 (ORT and CFIR). In general, combinations with two implementation TMFs received the most points. Two combinations each with two TMFs tied with the most points: (1) EPIS and Organizational Readiness Theory; (2) PRISM and CFIR.

The tie between the top two TMF combinations was broken through decision by research users from our partner LTC organization. They selected the PRISM and CFIR to collectively underpin our implementation objectives. Research user partners explained that their primary reason for choosing this combination was they had experience with the CFIR and appreciated the patient-centered focus of the PRISM, which aligned with the philosophy of care in our partner LTC organization.

## Discussion

Instead of relying on convenience or previous exposure, we developed a systematic, IKT and consensus-based method, referred to as the Implementation Theory Selection Model (ITSM),

**Table 5. Number of first, second and third place votes received for each potential combination of theories, models and/or frameworks (TMFs) in the third round of consensus of our case example, and the resulting points assigned to each combination.**

| Combination | No. Votes | | | Points |
|---|---|---|---|---|
| | 1st Place | 2nd Place | 3rd Place | |
| EPIS[a] + ORT[b] | 0 | 4 | 1 | 9 |
| PRISM[c] + CFIR[d] | 2 | 1 | 1 | 9 |
| EPIS[a] + CFIR[d] | 1 | 0 | 1 | 4 |
| PRISM[c] + ORT[b] | 1 | 0 | 2 | 5 |
| EPIS[a] + PRISM[c] + ORT[b] | 1 | 0 | 0 | 3 |
| EPIS[a] + ORT[b] + CFIR[d] | 1 | 1 | 0 | 5 |
| EPIS[a] + PRISM[c] + ORT[b] + CFIR[d] | 0 | 0 | 1 | 1 |

[a]EPIS = Exploration, Preparation and Sustainability Model.
[b]ORT = Organizational Readiness Theory.
[c]PRISM = Practical Robust Implementation and Sustainability Model.
[d]CFIR = Consolidated Framework for Implementation Research.

to guide the selection of TMFs for implementation processes. In a case example, we effectively applied the ITSM to select the PRISM and CFIR to collectively inform research on the implementation of hip protectors for fracture prevention in LTC.

The value of the ITSM is best illustrated by our selection of the PRISM, which no investigators or research user partners were aware of before undertaking this process. In fact, team members had experience with only two of the six TMFs short-listed for the consensus meeting. It is unlikely we would have chosen the PRISM to inform our implementation objectives had we not undertaken this process. Before this process, several team members had experience with the CFIR. AMBK and KMS used the CFIR to inform their hypotheses and to analyze data in other research studies. Research user partners (SB, CH) used the CFIR to inform past implementation efforts in our partner LTC organization. It is worth mentioning that after a very systematic and rigorous process, CFIR was chosen in the end by research users partly because of previous exposure, despite our explicit attempt to de-emphasize this criterion. This is consistent with findings from Birken et al. (2017), who found that familiarity was an important criteria for selecting implementation TMFs [8].

Application of the ITSM in our case example helped investigators and research users to improve their understanding and awareness of implementation TMFs, and helped to connect members of the research team by acting as a community-building activity. In particular, this allowed investigators to engage research users early in the project, after conception but before protocol development. Aligned with an IKT model of research collaboration, we relied on research users to break the tie between the top two combinations of TMFs that surfaced in the final round of the consensus meeting. Research users expressed that this made them feel valued and provided them with an opportunity to contribute to research decision-making in a very tangible way. It is anticipated that early engagement may help facilitate meaningful engagement in later stages or phases of the research project. We are currently engaging with research users who attended the consensus meeting on a subsequent project assessing the readiness of LTC homes within our partner organization to implement hip protectors.

Many questions about the acceptability, usability, appropriateness and ultimate impact of the ITSM remain unanswered. For example, does the ITSM meet the needs and expectations of implementation teams across a variety of implementation contexts? How practical is it for implementation teams to use the ITSM? What is the value-added of using the ITSM compared to traditional approaches to TMF selection? Does use of the ITSM result in the identification of different determinants of knowledge use, the selection of unique theory-based change methods and implementation strategies that more precisely address key determinants of change, and ultimately improved outcomes, when compared to "usual" methods for TMF selection? Such questions could be answered using methods adapted from ongoing studies (e.g. [50]) to examine the acceptability, usability, appropriateness and perceived impact of the ITSM.

We acknowledge several drawbacks to the ITSM and in particular, the extensive time required to undertake this method while there are calls for rapid, responsive, and relevant health research [51–54]. The feasibility of widescale use the ITSM, in its present form, may be limited. For investigators wishing to adopt a simplified version of the ITSM, we recommend restricting the search strategy to identify a smaller number of potentially relevant TMFs. All six TMFs that were shortlisted and advanced to step 5 are contained in the inventory of Implementation and Dissemination Models in Health Research and Practice (https://dissemination-implementation.org). This was a key resource for identifying TMFs. We also recommend defining stricter eligibility criteria for screening (step 2), so fewer TMFs advance to appraisal (step 3). As we estimate it took a minimum of one hour for one appraiser to appraise one TMF, the entire appraisal process was very labor-intensive (23 TMFs x 2 appraisers/TMF x 1 hour/appraiser = 46 person hours). We narrowed the pool of TMFs to only those with an

appropriate aim, but this could be narrowed further based on the socio-ecological level(s) included within the TMF, the inclusion and depth of analysis or operationalization of specific implementation constructs, and/or the orientation of the TMF based on the type of intervention and setting.

We encountered challenges engaging research users in the selection of guiding frameworks. We found it difficult to engage the older adult members of our community advisory board in the consensus meeting. Only one older adult member attended, who voted only in the first round of the meeting, and who did not participate in all rounds of discussion. After the meeting, several attendees in the consensus meeting commented that they valued the insight of our older adult partner, and how it swayed their own votes. It would have been even more difficult to engage residents of LTC, even though their insight would be so important. The challenges encountered likely resulted from our use of more consultative and extractive approaches to patient and public engagement, as well as insufficient efforts to address power imbalances between academic researchers, organizational partners and patient partners. For those wishing to engage patients and members of the public more meaningfully in the selection of theory, a more democratic, equitable approach to research 'co-production' is needed that prioritizes the unique and vital perspectives of patients over more conventional research methods and understandings of what constitutes 'academic' research [55].

Despite these limitations, we believe the ITSM offers a practical, step-by-step guide for implementation groups to adopt a rigorous, transparent and reproducible method for TMF selection. It incorporates best practices for the selection of theory in implementation research and practice, by relying on existing tools (e.g., T-CaST, inventory of Implementation and Dissemination Models in Health Research and Practice), adhering to existing recommendations, and incorporating the feedback and advice of research users when reaching consensus on the final choice of TMFs. Although we have demonstrated the feasibility of operationalizing each step of the ITSM in our case example, continued research is needed to evaluate and refine the ITSM to ensure it is appropriate for a wide variety of implementation contexts.

## Conclusions

We developed a systematic, consensus-based method for identifying and selecting TMFs, referred to as the **I**mplementation **T**heory **S**election **M**odel (ITSM). The ITSM comprises five steps: (step 1) identify potentially relevant TMFs; (step 2) narrow the pool of TMFs; (step 3) appraise eligible TMFs; (step 4) prioritize a short-list of TMFs for in-depth consideration; and (step 5) select TMFs through consensus with investigators and research user partners. We then applied the ITSM to choose TMFs to inform research on the implementation of hip protectors for fracture prevention in LTC. This resulted in the selection of the PRISM and CFIR. The value of the ITSM is best reflected by our selection of the PRISM, which no investigators and researcher users were familiar with before undertaking this process. The ITSM offers a practical, step-by-step guide for implementation groups to adopt a rigorous, transparent and reproducible method for TMF selection, although continued research is needed to evaluate and refine this method to ensure it is appropriate for a wide variety of implementation contexts.

## Supporting information

**S1 File. Theory, Model and/or Framework (TMF) appraisal form.**
(PDF)

**S2 File. Example summary statement of the Practical, Robust Implementation and Sustainability Model (PRISM).**
(PDF)

**S3 File. Agenda for the consensus meeting in our case example.**
(PDF)

**S4 File. List of included and excluded theories, models and/or frameworks (TMFs) in our case example.**
(PDF)

**S5 File. Categorized aims of included and excluded theories, models and/or frameworks (TMFs) in our case example.**
(XLSX)

**S6 File. Appraisal scores of eligible theories, models and/or frameworks (TMFs) in our case example.**
(XLSX)

## Acknowledgments

We express our gratitude to Alexie Touchette for assisting with the screening of theories, models and/or frameworks; to Devashree Prahbu for helping with the conduct of the consensus meeting; to Anne Kerr for engaging in the consensus meeting; and to Dawn Steliga for assisting with data entry.

## Author Contributions

**Conceptualization:** Alexandra M. B. Korall, Helen Chong, Vicki Komisar, Dawn C. Mackey, Masood Khan, Femke Hoekstra, Susan G. Brown, Pauli Gardner, Christine Hames, Andrew C. Laing, Kathryn M. Sibley.

**Data curation:** Alexandra M. B. Korall, Kathryn M. Sibley.

**Formal analysis:** Alexandra M. B. Korall, Kathryn M. Sibley.

**Funding acquisition:** Kathryn M. Sibley.

**Investigation:** Alexandra M. B. Korall, Kathryn M. Sibley.

**Methodology:** Alexandra M. B. Korall, Kathryn M. Sibley.

**Project administration:** Kathryn M. Sibley.

**Resources:** Alexandra M. B. Korall, Kathryn M. Sibley.

**Software:** Kathryn M. Sibley.

**Supervision:** Alexandra M. B. Korall, Kathryn M. Sibley.

**Validation:** Alexandra M. B. Korall, Kathryn M. Sibley.

**Visualization:** Alexandra M. B. Korall, Kathryn M. Sibley.

**Writing – original draft:** Alexandra M. B. Korall, Kathryn M. Sibley.

**Writing – review & editing:** Alexandra M. B. Korall, Kathryn M. Sibley.

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
