## [Decision Letter · Decision Letter 0]

29 Apr 2024

PONE-D-24-05799Proposal of the Implementation Theory Selection Model and exemplar application in fall injury prevention: A systematic, consensus-based method to select implementation theories, models and/or frameworksPLOS ONE

Dear Dr. Sibley,

Thank you for submitting your manuscript to PLOS ONE. After careful consideration, we feel that it has merit but does not fully meet PLOS ONE’s publication criteria as it currently stands. Therefore, we invite you to submit a revised version of the manuscript that addresses the points raised during the review process.

We look forward to receiving your revised manuscript.

Kind regards,

Ivan Sarmiento

Academic Editor

PLOS ONE

Journal Requirements:

2. In the online submission form, you indicated that "The data underlying the results presented in the study may be available from the corresponding author on request."

6. Please include your tables as part of your main manuscript and remove the individual files. Please note that supplementary tables (should remain/ be uploaded) as separate ""supporting information"" files

Additional Editor Comments:

I used the major revision option to give you more time to review, but I hope you will find it possible to address the comments in a shorter time.==============================

Reviewers' comments:

Reviewer's Responses to Questions

**Comments to the Author**

1. Is the manuscript technically sound, and do the data support the conclusions?

Reviewer #1: Yes

Reviewer #2: Yes

2. Has the statistical analysis been performed appropriately and rigorously? 

Reviewer #1: N/A

Reviewer #2: N/A

3. Have the authors made all data underlying the findings in their manuscript fully available?

Reviewer #1: Yes

Reviewer #2: Yes

4. Is the manuscript presented in an intelligible fashion and written in standard English?

Reviewer #1: Yes

Reviewer #2: Yes

5. Review Comments to the Author

Reviewer #1: Dear co-authors of the manuscript "Proposal of the Implementation Theory Selection Model and exemplar application in fall injury prevention",

Thank you for the opportunity to review the above manuscript and for your patience in receiving feedback. I enjoyed reviewing this submission, which I view as a helpful example for how to collaborate with research users in designing a study (we know far too little about this), and in particular selecting guiding frameworks. Below, I share a number of reflections, questions and comments that I think could help to further improve your submission, which I would be happy to review again. My thoughts are listed chronologically, following the manuscript page by page.

(1) Page 1, line 19: an "r" is missing in "Schlegel-University"

(2) Page 6, line 110: I appreciate that you mention the T-CaSt here and make it clear where your approach adds value.

(3) Page 8/9, line 165/ 172: Here you highlight that it is important to find sources that fit the implementation context and/or stem from the same or a similar context as the implementation in focus. While this makes sense intuitively, I still wonder why this is important - a clearer argument here would be desirable. Or, said in a different way, why would a framework developed in education be less usable than one developed in the field of chronic diseases? I think it is worth making this more explicit. Additionally, I wondered how easy (or difficult) it was for you to make this assessment, and which criteria were used to conclude, "okay, this is close enough to our context". This may be hidden somewhere in the supplementary materials - and could be worth a remark or two in the text.

(4) Page 8, line 180: A reader may be surprised about the fact that you sat with a review of TMFs used in prevention and management of falls in older adults - and nevertheless looked beyond this review for potentially usable TMFs that could fit your needs. Why did you feel this was worth going through? You describe a lot of complex, laborious legwork, and I think it is important to make it very clear why it was worth it for you.

(5) Page 10, line 193: You describe how you could make decisions on which TMFs to in-/exclude based on different criteria. Is your experience that TMFs are sufficiently clear on these criteria such that these decisions are easy to make? This may be worth a remark here but could also be part of the discussion.

(6) In this discussion, I am missing reflections on challenges to the process of selecting guiding frameworks in a collaboration between researchers and research users, including, for example, reflections on positionality, power imbalances and other factors that have been highlighted in the literature as the "dark sides" of co-production (see, e.g., Oliver et al., 2019 or Williams et al. 2020; Boaz et al., 2021). I am aware that you describe some - e.g., engagement - challenges but I see only scarce information on how you view your own role in the process described and which challenges such a process potentially presents for research processes.

(7) Finally, I wondered if a simple figure of the ITSM could be added to the manuscript itself. I am aware that you have Table 1 in the supplementary materials, but nevertheless felt it could be helpful to have the five steps represented in a simple graphic that could go directly into the manuscript.

Reviewer #2: Thank you for the opportunity to review PONE-D-24-05799, “Proposal of the Implementation Theory Selection Model and exemplar application in fall injury prevention: A systematic, consensus-based method to select implementation theories, models and/or frameworks.” I believe that the paper fills a critical gap in the TMF selection space by offering and demonstrating practical steps. The paper is well-written, concise, and instructive. I particularly like the tables and additional files; I imagine that they’ll be very useful to investigators looking to use ITSM. I have just three concerns that I hope the authors can address:

First, although the authors’ application of the ITSM was highly rigorous, they do not explain how they developed the model. It is unclear how they arrived at the model’s five steps and sub-steps that articulate how to carry out each step. Related, some suggestions for sub-steps seem overly specific (e.g., in step 1, why only include papers that were cited at least 10 times?), and I was left wondering how the authors arrived at these criteria. A clear description of the methods by which they developed the model would be useful, maybe in the ‘overview of the model’ section on page 8.

Second, I was glad that the discussion section acknowledged the large number and amount of resources required (e.g., at least two reviewers screen each TMF) to use the model as proposed, and I liked the suggestions that the authors offered for simplifying the steps on p 21-22 are useful, but I imagine that ITSM will be most useful to solo investigators who may not have the resources to carry out even the simplified approach. The paper would be strengthened with recommendations for an individual without the significant resources that the author had. I don’t think they need to endorse such a bare bones approach, but they could acknowledge the need for one, and propose modifications that would make it feasible for someone with few resources (including a deep bench of collaborators).

Third, in the discussion section on page 21, the authors acknowledge that the influence of the model is unclear. This is a point worth fleshing out a bit more. There are a lot of questions – about usability of the model, the value-add beyond traditional approaches to TMF selection, etc. In addition, how the COAST-IS trial would inform the model’s development was not clear. More attention should be paid to what a future research agenda should look like in this respect.

6. PLOS authors have the option to publish the peer review history of their article (what does this mean?). If published, this will include your full peer review and any attached files.

Reviewer #1: **Yes: **Bianca Albers

Reviewer #2: **Yes: **Sarah Abigail Birken

---

## [Author Response · Author response to Decision Letter 0]

15 Jul 2024

RESPONSE TO REVIEWERS

MS. Ref. No.: PONE-D-24-05799

Title: Proposal of the Implementation Theory Selection Model and exemplar application in fall injury prevention

Comments from the Editor:

Response: We have made the formatting of our manuscript compliant with PLOS ONE’s style requirements.

2. In the online submission form, you indicated that "The data underlying the results presented in the study may be available from the corresponding author on request."

Response: We have uploaded the datasets supporting our findings in the supplementary information files S5 and S6.

Response: We have amended the title on the online submission form to match the title in the manuscript.

Response: Research users were not human participants in the research process, but instead were partners on the research team (level of ‘consult’ on the IAP2 Spectrum of Patient and Researcher Engagement in Health Research). Given the collaborative role of research users in the research process and as is common practice in collaborative methods of health research, research ethics board (REB) approval was not sought, and informed written or verbal consent to participate was not obtained for the involvement of research users in this capacity. To clarify, we have added the following statement to Page 6, Lines 132-138 of the Methods section (new text shown in green highlighter; line numbers correspond to the file labeled Revised Manuscript with Track Changes):

“Research user partners were not human participants in the research process, but instead were partners on the research team (level of ‘consult’ on the IAP2 Spectrum of Patient and Researcher Engagement in Health Research). Given the collaborative role of research users in the research process and as is common practice in collaborative methods of health research, research ethics board (REB) approval was not sought, and informed written or verbal consent to participate was not obtained for the involvement of research users in this capacity.”

Response: We have included a separate caption for each figure in the manuscript.

6. Please include your tables as part of your main manuscript and remove the individual files. Please note that supplementary tables (should remain/ be uploaded) as separate "supporting information" files.

Response: We have included tables as part of the main manuscript and removed the individual files. We have uploaded separate “supporting information” files for supplementary files.

Comments from Reviewer #1:

Dear co-authors of the manuscript "Proposal of the Implementation Theory Selection Model and exemplar application in fall injury prevention", Thank you for the opportunity to review the above manuscript and for your patience in receiving feedback. I enjoyed reviewing this submission, which I view as a helpful example for how to collaborate with research users in designing a study (we know far too little about this), and in particular selecting guiding frameworks. Below, I share a number of reflections, questions and comments that I think could help to further improve your submission, which I would be happy to review again. My thoughts are listed chronologically, following the manuscript page by page.

Response: Thank you for your positive appraisal of our manuscript. We hope the changes made to the manuscript have addressed your reflections, comments and questions.

1. Page 1, line 19: an "r" is missing in "Schlegel-University"

Response: We have corrected this mistake.

2. Page 6, line 110: I appreciate that you mention the T-CaSt here and make it clear where your approach adds value.

Response: Thank you for your positive comment. No changes were made.

3. Page 8/9, line 165/ 172: Here you highlight that it is important to find sources that fit the implementation context and/or stem from the same or a similar context as the implementation in focus. While this makes sense intuitively, I still wonder why this is important - a clearer argument here would be desirable. Or, said in a different way, why would a framework developed in education be less usable than one developed in the field of chronic diseases? I think it is worth making this more explicit. Additionally, I wondered how easy (or difficult) it was for you to make this assessment, and which criteria were used to conclude, "okay, this is close enough to our context". This may be hidden somewhere in the supplementary materials - and could be worth a remark or two in the text.

Response: We included this statement in recognition of the fact that considering TMFs that have been used in similar contexts may serve as a useful starting point for considering. However, we also note the importance of not restricting consideration in this way (which is consistent with the Reviewer’s example above). We have removed this recommendation and now advocate against being too prescription when identifying a pool of potentially relevant TMFs. To address this comment, we removed the sentence on Page 9, Line 169 that read “The sources searched in this step should be adapted to fit the implementation context.” We also modified (new text shown in green highlighter) the following sentences on Pages 9-10, Lines 174-180 to read:

Complementary sources may include published reviews and academic articles identifying TMFs used in implementation research and practice, ideally including some conducted with similar target populations, interventions or settings, as these characteristics are relatively easy to assess. In general, we advise against being overly prescriptive when developing a list of candidate TMFs, as the association between characteristics of a TMF and its ultimate relevance will be unclear at this step.

4. Page 8, line 180: A reader may be surprised about the fact that you sat with a review of TMFs used in prevention and management of falls in older adults - and nevertheless looked beyond this review for potentially usable TMFs that could fit your needs. Why did you feel this was worth going through? You describe a lot of complex, laborious legwork, and I think it is important to make it very clear why it was worth it for you.

Response: In line with our response to the previous comment, and in acknowledging the Reviewer’s example, we do feel it is important to not restrict initial considerations just to previous applications in a specific area. To justify our approach, we added the following sentence to Page 10, Lines 192-194:

“We elected to look beyond this review of implementation research on falls in older adults to compile a comprehensive list of candidate TMFs.”

5. Page 10, line 193: You describe how you could make decisions on which TMFs to in/exclude based on different criteria. Is your experience that TMFs are sufficiently clear on these criteria such that these decisions are easy to make? This may be worth a remark here but could also be part of the discussion.

Response: Based on our experience, it is not always easy to make decisions regarding eligibility of TMFs. Hence our recommendation that each TMF be screened by at least two independent reviewers, with discussions resolved through discussion or a third independent reviewer. To emphasize our rationale for using this approach, we have made the following edits (changes shown in green highlighter) to the following text on Page 11 Lines 209-213:

“As it can be somewhat challenging to make decisions regarding the eligibility of TMFs, ideally, at least two reviewers should independently screen each TMF, and discrepancies should be resolved through discussion or a third independent reviewer. When a complete dual review process is not feasible, single screening should be conducted only by an experienced reviewer (40).” 

6. In this discussion, I am missing reflections on challenges to the process of selecting guiding frameworks in a collaboration between researchers and research users, including, for example, reflections on positionality, power imbalances and other factors that have been highlighted in the literature as the "dark sides" of co-production (see, e.g., Oliver et al., 2019 or Williams et al. 2020; Boaz et al., 2021). I am aware that you describe some - e.g., engagement - challenges but I see only scarce information on how you view your own role in the process described and which challenges such a process potentially presents for research processes.

Response: Thank you for your insightful comment and for providing these useful references. We have expanded our discussion to reflect on positionality, power imbalances and our more consultative, extractive approach to engaging patients and members of the public. The discussion section now includes the following paragraph (new text highlighted in green) on Pages 28-29, Lines 511-524:

“We encountered challenges engaging knowledge users in the selection of guiding frameworks. We found it difficult to engage the older adult members of our community advisory board in the consensus meeting. Only one older adult member attended, who voted only in the first round of the meeting, and who did not participate in all rounds of discussion. After the meeting, several attendees in the consensus meeting commented that they valued the insight of our older adult partner, and how it swayed their own votes. It would have been even more difficult to engage residents of LTC, even though their insight would be so important. The challenges encountered likely resulted from our use of more consultative and extractive approaches to patient and public engagement, as well as insufficient efforts to address power imbalances between academic researchers, organizational partners and patient partners. For those wishing to engage patients and members of the public more meaningfully in the selection of theory, a more democratic, equitable approach to research ‘co-production’ is needed that prioritizes the unique and vital perspectives of patients over more conventional research methods and understandings of what constitutes ‘academic’ research (55).”

7. Finally, I wondered if a simple figure of the ITSM could be added to the manuscript itself. I am aware that you have Table 1 in the supplementary materials, but nevertheless felt it could be helpful to have the five steps represented in a simple graphic that could go directly into the manuscript.

Response: We have added a figure of the ITSM (Fig 1).

Comments from Reviewer #2:

Thank you for the opportunity to review PONE-D-24-05799, “Proposal of the Implementation Theory Selection Model and exemplar application in fall injury prevention: A systematic, consensus-based method to select implementation theories, models and/or frameworks.” I believe that the paper fills a critical gap in the TMF selection space by offering and demonstrating practical steps. The paper is well-written, concise, and instructive. I particularly like the tables and additional files; I imagine that they’ll be very useful to investigators looking to use ITSM. I have just three concerns that I hope the authors can address:

Response: We thank the reviewer for their positive critique of our manuscript.

1. First, although the authors’ application of the ITSM was highly rigorous, they do not explain how they developed the model. It is unclear how they arrived at the model’s five steps and sub-steps that articulate how to carry out each step. Related, some suggestions for sub-steps seem overly specific (e.g., in step 1, why only include papers that were cited at least 10 times?), and I was left wondering how the authors arrived at these criteria. A clear description of the methods by which they developed the model would be useful, maybe in the ‘overview of the model’ section on page 8.

Response: We agree with the reviewer that the manuscript was lacking details on how we developed the five steps of the ITSM and the sub-steps we used in our example to operationalize each step of the ITSM. However, we want to clarify that we are not advocating for others to use the same sub-steps as we did in our case example, but instead that each research team should work collaboratively with knowledge users to develop the specific sub-steps they will use to operationalize the five steps of the ITSM. We have added the following text (new text shown in green highlighter) to the section “Overview of the Implementation Theory Selection Model” on Page 7, Lines 147-159 to describe these details:

“The ITSM was informed by the JLA method for priority setting (https://www.jla.nihr.ac.uk/jla-guidebook/). The JLA method includes steps for gathering data on potential research questions, evidence checking, interim priority setting, and the final prioritization of research questions through a workshop. AMBK and KMS proposed a first iteration of the ITSM’s five steps, which were revised through consultation with our academic and organizational team members. Once the five steps of the ITSM were finalized, AMBK and KMS proposed the sub-steps to carry out each step of the ITSM in our case example on hip protector implementation, which was revised iteratively through discussions with our research team as the process evolved. In the sections below, we describe and provide examples of how we operationalized each step of the ITSM in our case example on hip protector implementation to illustrate the feasibility and potential value of the ITSM. We encourage ITSM users to work collaboratively with research user partners to develop the specific sub-steps they will use to operationalize the ITSM’s five steps which are feasible given their access to available resources.

We have also added the following sentences to Page 10, Lines 188-190 describing our specific rationale for only including TMFs with at least 10 citations from the scoping review on chronic disease management and prevention in Step 1 of our case example:

“Our decision to include TMFs with ten or more citations from the scoping review was pragmatic; our goal was to ensure we weren’t missing TMFs that are commonly used in a related field of interest.”

2. Second, I was glad that the discussion section acknowledged the large number and amount of resources required (e.g., at least two reviewers screen each TMF) to use the model as proposed, and I liked the suggestions that the authors offered for simplifying the steps on p 21-22 are useful, but I imagine that ITSM will be most useful to solo investigators who may not have the resources to carry out even the simplified approach. The paper would be strengthened with recommendations for an individual without the significant resources that the author had. I don’t think they need to endorse such a bare bones approach, but they could acknowledge the need for one, and propose modifications that would make it feasible for someone with few r

---

## [Decision Letter · Decision Letter 1]

15 Aug 2024

PONE-D-24-05799R1Proposal of the Implementation Theory Selection Model and exemplar application in fall injury preventionPLOS ONE

Dear Dr. Sibley,

Thank you for submitting your manuscript to PLOS ONE. After careful consideration, we feel that it has merit but does not fully meet PLOS ONE’s publication criteria as it currently stands. Therefore, we invite you to submit a revised version of the manuscript that addresses the points raised during the review process.

Please address the comment from reviewer #2 regarding the relevance of the reference to COAST-IS. The reviewer has made a valid point that should be taken into consideration.

The authors should provide a reference to support the statement on page 6, line 133, about the need for an ethics review. Is the collaborative nature of the research the reason for not needing a review, or is it the nature of the study and the source of data? This statement should be in line with relevant guidelines for the research context such as the TCP2.

“Given the collaborative role of research users in the research process and as is common practice in collaborative methods of health research, research ethics board (REB) approval was not sought, and informed written or verbal consent to participate was not obtained for the involvement of research users in this capacity.”

We look forward to receiving your revised manuscript.

Kind regards,

Ivan Sarmiento

Academic Editor

PLOS ONE

Journal Requirements:

Additional Editor Comments:

Please address the comment from reviewer #2 regarding the relevance of the reference to COAST-IS. The reviewer has made a valid point that should be taken into consideration.

The authors should provide a reference to support the statement on page 6, line 133, about the need for an ethics review. Is the collaborative nature of the research the reason for not needing a review, or is it the nature of the study and the source of data? This statement should be in line with relevant guidelines for the research context such as the TCP2.

“Given the collaborative role of research users in the research process and as is common practice in collaborative methods of health research, research ethics board (REB) approval was not sought, and informed written or verbal consent to participate was not obtained for the involvement of research users in this capacity.” (p6, l133)

Reviewers' comments:

Reviewer's Responses to Questions

**Comments to the Author**

1. If the authors have adequately addressed your comments raised in a previous round of review and you feel that this manuscript is now acceptable for publication, you may indicate that here to bypass the “Comments to the Author” section, enter your conflict of interest statement in the “Confidential to Editor” section, and submit your "Accept" recommendation.

Reviewer #1: All comments have been addressed

Reviewer #2: (No Response)

2. Is the manuscript technically sound, and do the data support the conclusions?

Reviewer #1: Yes

Reviewer #2: Yes

3. Has the statistical analysis been performed appropriately and rigorously? 

Reviewer #1: N/A

Reviewer #2: N/A

4. Have the authors made all data underlying the findings in their manuscript fully available?

Reviewer #1: Yes

Reviewer #2: Yes

5. Is the manuscript presented in an intelligible fashion and written in standard English?

Reviewer #1: Yes

Reviewer #2: Yes

6. Review Comments to the Author

Reviewer #1: Thank you for having addressed al my comments - and for being responsive to suggestions made by another reviewer, which I also found helpful for revising the manuscript. I have nothing further to suggest and look forward to seeing this published.

Reviewer #2: Thank you for the thorough responses to my initial review. I remain confused about the reference to COAST-IS. I think that the authors may be suggesting that the cited protocol for a pilot, which tees up a comparison of COAST-IS (an implementation strategy selection and tailoring approach) and usual care makes for a good template for comparison of ITSM (a TMF selection approach) to usual care. There are so many pilots intended to tee up a comparison of a new approach to usual care; I don't understand why COAST-IS, a protocol published more than 4 years ago, for which there are no subsequent publications and therefore has ostensibly been abandoned by the study team, is used to make this point. It feels like a red herring that could be made without reference to any other approach, just by saying that comparing ITSM to standard approaches of TMF selection are warranted. As currently written, I find the reference to COAST-IS to distract from what is a pretty straightforward point.

7. PLOS authors have the option to publish the peer review history of their article (what does this mean?). If published, this will include your full peer review and any attached files.

Reviewer #1: **Yes: **Bianca Albers

Reviewer #2: **Yes: **Sarah A. Birken

---

## [Author Response · Author response to Decision Letter 1]

20 Aug 2024

Author response:

Thank you to the reviewers and editors for reviewing our manuscript. We have responded to the last two outstanding comments below and made relevant changes to the manuscript. We note the reviewers were otherwise satisfied with the revisions.

Associate Editor/ Reviewer #2 comment:

Please address the comment from reviewer #2 regarding the relevance of the reference to COAST-IS. The reviewer has made a valid point that should be taken into consideration. “Thank you for the thorough responses to my initial review. I remain confused about the reference to COAST-IS. I think that the authors may be suggesting that the cited protocol for a pilot, which tees up a comparison of COAST-IS (an implementation strategy selection and tailoring approach) and usual care makes for a good template for comparison of ITSM (a TMF selection approach) to usual care. There are so many pilots intended to tee up a comparison of a new approach to usual care; I don't understand why COAST-IS, a protocol published more than 4 years ago, for which there are no subsequent publications and therefore has ostensibly been abandoned by the study team, is used to make this point. It feels like a red herring that could be made without reference to any other approach, just by saying that comparing ITSM to standard approaches of TMF selection are warranted. As currently written, I find the reference to COAST-IS to distract from what is a pretty straightforward point.”

Response: We have revised the statements in this section which reduce emphasis on the COAST-IS trial. We have made the statements more general and offer the reference to COAST-IS as an example. The pertinent section now refers to lines 460-472 and reads: “Such questions could be answered using methods adapted from ongoing studies (e.g. [50]) to examine the acceptability, usability, appropriateness and perceived impact of the ITSM.”

Associate Editor comment:

The authors should provide a reference to support the statement on page 6, line 133, about the need for an ethics review. Is the collaborative nature of the research the reason for not needing a review, or is it the nature of the study and the source of data? This statement should be in line with relevant guidelines for the research context such as the TCP2: “Given the collaborative role of research users in the research process and as is common practice in collaborative methods of health research, research ethics board (REB) approval was not sought, and informed written or verbal consent to participate was not obtained for the involvement of research users in this capacity.”

Response: We have revised this text and offered a reference in support of the practice of not seeking research ethics approval for individuals involved in research decision making such as theory selection. This section (lines 131-137 now read: “The individuals involved in the theory selection process were not human research participants, but instead were partners on the research team (level of ‘consult’ on the IAP2 Spectrum of Patient and Researcher Engagement in Health Research). Given the collaborative decision making role of these partners in the research process, research ethics board (REB) approval was not sought. This practice was consistent with the JLA guidance that views such decision making work as service evaluation and development (https://www.jla.nihr.ac.uk/jla-guidebook/).”

---

## [Editor Report · Decision Letter 2]

26 Aug 2024

Proposal of the Implementation Theory Selection Model and exemplar application in fall injury prevention

PONE-D-24-05799R2

Dear Dr. Sibley,

We’re pleased to inform you that your manuscript has been judged scientifically suitable for publication and will be formally accepted for publication once it meets all outstanding technical requirements.

Kind regards,

Ivan Sarmiento

Academic Editor

PLOS ONE
---

## [Editor Report · Acceptance letter]

2 Sep 2024

PONE-D-24-05799R2 

PLOS ONE

Dear Dr. Sibley, 

I'm pleased to inform you that your manuscript has been deemed suitable for publication in PLOS ONE. Congratulations! Your manuscript is now being handed over to our production team.

Kind regards, 

on behalf of

Dr. Ivan Sarmiento 

Academic Editor

PLOS ONE